

# Not always a matter of context: direct effects of red on arousal but context-dependent moderations on valence

Vanessa L. Buechner and Markus A. Maier

Department of Psychology, Ludwig-Maximilians-Universität München, Munich, Germany

## ABSTRACT

The arousal theory of color proposes that red is associated with arousal. Research on the color-in-context theory, in turn, states that the context in which red is perceived influences its valence-related meaning and behavioral responses to it. This study faces and integrates these theories by examining the influence of red on both arousal and valence perceptions of test-relevant and neutral stimuli, rendering a color 2 (red vs. blue) × context 2 (test vs. neutral) between-subjects design. Participants rated different pictures regarding their arousal and valence component, respectively. In line with the assumptions of both theories, red increased arousal perceptions of stimuli irrespective of their valence but a context × color interaction was found for valence perceptions: for participants viewing test-relevant pictures, red increased their perceptions of negativity compared to neutral pictures. The present study shows that both theories are actually compatible when differentiating the arousal and valence component.

# INTRODUCTION

Research on the meaning and effects of colors has been of interest since the 1940's. Thereby, two different theories primarily focusing on effects of the color red met approval in the last years—the *arousal theory of color* and the *color-in-context theory*. As their names imply, the arousal theory states that the color red per se is arousing, whereas the color-in-context theory states that context in which this color is perceived influences its valence-related meaning and behavioral responses to it. Plenty of research has been done on both theories and will be briefly described in the following paragraph.

In 1942 Goldstein stated that color perception produces physiological reactions in the body that are manifest in emotions, cognitions, and behavior (*Goldstein, 1942*). Based on this idea, later color researchers posited that longer wavelength colors, such as red, are experienced as arousing and stimulating, whereas shorter wavelength colors, such as green and blue, are relaxing (e.g., *Crowley, 1993*). *Nakshian (1964)* provided support for the notion that red is physiologically exciting or arousing, as red, in comparison to green, leads to greater hand tremor and faster speed of movement in a motor inhibition task. Similarly, *Gerard (1958)* showed that the perception of red light leads to an increase of

Corresponding author
Vanessa L. Buechner,
vanessa.buechner@psy.lmu.de

blood pressure, respiratory movements, and frequency of eye blink, whereas blue light leads to a decrease of these parameters. Interestingly, *Wilson (1966)* showed that red, in comparison to green, is not only more arousing from an implicit physiological (i.e., high skin conductance level and response) but also from a more explicit cognitive (i.e., ratings) perspective. Using Likert scales research could show that the color red is associated with terms such as 'exciting' and 'stimulating' (*Wexner, 1954*) as well as 'warmth' and 'less calmness' (*Wright & Rainwater, 1962*).

Taken together, previous research using both explicit and implicit measurement tools provided evidence that the color red, as a typical long-wavelength color, is correlated with high arousal whereas short-wavelength colors reduce the individual's arousal level.

The second theory, the color-in-context theory (*Elliot & Maier, 2012*) focuses on the valence dimension of colors and, by drawing on biology and social learning, claims that the context in which color is perceived influences its affective meaning and responses to it. Context is defined as both the physical structure as well as the psychological situation (*Bazire & Brézillon, 2005*; *Zimmermann, Lorenz & Oppermann, 2007*) within which color is embedded. These circumstances frame a color and determine its meaning. More precisely, with regard to the color red in achievement contexts this color becomes associated with both 'danger' and 'caution' (*Moller, Elliot & Maier, 2009*) and therefore activates avoidance motivation (*Elliot et al., 2009*; *Maier, Elliot & Lichtenfeld, 2008*). In romantic contexts, however, red becomes associated with 'sex' and 'romance' and consequently triggers approach motivation (*Elliot et al., 2010*; *Kaya & Epps, 2004*; *Niesta Kayser, Elliot & Feltman, 2010*; *Meier et al., 2012*). Thus, depending on the situational circumstances red seems to take on opposite meanings on the valence dimension (see also *Maier et al., 2009*).

The original color-in-context theory favored an association approach assuming that red is associated with positive valence and approach tendencies in one context (e.g., dating) and negative valence and avoidance tendencies within other contextual circumstances (e.g., achievement or threat to life). However, this model has recently been refined with regard to the exact nature of how the color red interacts with the context. Instead of the activation of different associations in the presence of red through various contexts, red was proposed to exert an attentional bias towards the object that is displaying the color red. More precisely, Buechner and colleagues could show that red actually functions as a signal of importance, which carries the information that a stimulus is worthy of attention. This attentional bias in turn emphasizes a stimulus' existing motivational message and consequently increases the perceivers' appropriate -that is valence dependent- response tendencies: increased approach in case of a positive-appetitive stimulus configuration and increased avoidance in case of a negative-aversive one (*Buechner et al., 2014*; *Buechner et al., 2015*). This was shown in a study using a head and upper torso photo of a young man wearing a white shirt. Color was manipulated by placing a red or blue circle on the left chest of the target male and the type of facial expression and body posture (i.e., proud and ashamed male; see University of California-Davis Set of Emotion Expressions; *Tracy, Robins & Schriber, 2009*) constituted the context. The results showed that in female mate evaluation, red increases the perceived attractiveness of a proud man (i.e., an appetitive signal of mate value) but tends to decrease the perceived attractiveness of an ashamed man

(i.e., an aversive signal of low mate value). To sum up, the updated color-in-context theory claims that red accentuates a stimulus' affective meaning and subsequently defines or triggers the corresponding valence of this stimulus. Thus, the proposed impact color might exert in the present study is different from previous studies: it slightly intensifies the meaning of the negative test items, but does not change it. Importantly, as predicted, red effects that vary with the context have only been found on the valence dimension of the stimuli used.

In light of both theories, red is expected to increase arousal regardless of contextual variations in which the color is perceived, but red is supposed to trigger the perception of valence along with and therefore dependent on the context-related information being present.

### Aims and hypotheses of the present research

This study aims to examine both the arousal and the color-in-context hypothesis by exploring the influence of colored stimuli on arousal and valence perceptions. The study described herein is thus a conceptual replication of previous findings documenting the intensifying effect of the color red on valence perceptions (*Buechner et al., 2015*) with a highly powered sample in another context, that is achievement context instead of romantic. Further, experimental studies often used picture stimuli to create romantic settings, but never to create achievement contexts. Achievement contexts were mostly induced via instructions or specific performance tasks. Here we fill this gap by using test-relevant and neutral pictures to create an achievement context. That is, a set of test-relevant and neutral pictures will be presented together with a red or blue frame. Red should raise arousal perceptions for both picture types thus confirming the arousal theory but it should increase valence perceptions only for test-relevant pictures confirming the color-in-context theory. [1]

## MATERIALS & METHODS

In line with *Simmons, Nelson, and Simonsohn*'s *(2012)* proposal for a 21 word solution of disclosure, we report determination of sample size, all data exclusions, all manipulations, and all measures in the study.

### Participants

Before the beginning of the study we performed an a priori power analysis using G*Power 3.1.3 (*Faul et al., 2007*). Given a color 2 (red vs. blue) × context 2 (test vs. neutral) between-subjects design with four separate experimental groups, the analysis suggested a sample size of 309 participants to provide 80% power at $\alpha = 0.05$ assuming a small effect size ($d = .16$[2]). Thus, a total of 310 undergraduate students at German universities participated in the study, 71 in the red test condition, 77 in the red neutral condition, 81 in the blue test condition, and 76 in the blue neutral condition. Five participants were excluded due to color blindness, resulting in a final sample of 305 (74 males, *mean age =* 22.28 years, $SD = 5.52$) undergraduate students. This is the only study on color and arousal and valence perceptions that we have conducted.

[1]Someone could argue that the research question is redundant. The arousal theory and the color-in-context theory are based on very different levels of processing. Specifically, the arousal theory posits that red deserves its meaning from its wavelength characteristics, while the color-in-context theory focuses on the affective meaning of red on a more cognitive level. From this perspective, it seems logical that the associated arousal does not change with context. However, although theoretically convincing, empirical evidence is missing. Red effects on arousal have never been studied within an experimental context manipulation. Hence, although the main effect of color red on arousal was widely accepted potential moderating influences were never empirically ruled out. (We would like to thank reviewer 1 for this suggestion).

[2]The effect size of a similar study (*Buechner et al, 2015*) was small ($d = .20$). To be more conservative, we assumed a smaller effect size.

## Design and procedure

Participants were randomly assigned to one of the four experimental conditions, rendering a color 2 (red vs. blue) × context 2 (test vs. neutral) between-subjects design. The experimenters were blind to condition and hypotheses. Participants were instructed to open a folder to view 4 different pictures, around 6.7-in. × 4.5-in. in size and printed with an Epson Stylus Photo R800 color printer on a white heavyweight Epson Matte Paper, 8.2-in. × 11.6-in. in size. We handed four pictures to enable enough stimuli variability and to reduce the likelihood of unknown confounds. More precisely, dependent on the experimental group, they either saw four red-primed test-relevant pictures, 4 blue-primed test-relevant pictures, four red-primed neutral pictures, or four blue-primed neutral pictures. Three neutral pictures were taken from the International Affective Picture System (IAPS; *Lang, Bradley & Cuthbert, 2008*), which provides an experimental set of 1,169 digitized photographs with normative rating scores (using a 9-point rating scale) on valence and arousal. The neutral pictures were selected based on the normative valence ratings and showed scenarios, such as a theater hall (picture #2037), a newspaper (#7061), or a person reading a book (picture #2377). The forth neutral picture showing a stage was retrieved from the internet and judged as neutral by two dependent experts beforehand. Content-matched negative, test-relevant pictures, retrieved from the internet and judged as negative by two dependent experts beforehand, showed scenarios, such as a lecture hall while students writing an exam, a person being frustrated while working on a paper, a person being frustrated while reading a book, or a lecture hall. Color was manipulated by encircling the pictures with a red or blue frame, 0.3-in. × 0.3-in. in size. A spectrophotometer was used to determine the precise parameters of the colors: red (LCh[55,5; 52,2; 32,0]) and blue (LCh[55,2; 52,0; 274,7]); lightness and chroma were equated across hue. After participants viewed each of the respective four photos (dependent of condition) for five seconds, they were instructed to complete the questionnaire containing the dependent measures of both arousal and valence, demographic items, as well as some control items (i.e., trait and test anxiety) to test for any moderating effects. Someone could assume that high anxious individuals might exhibit a stronger color × context on valence effect. No specific effects on arousal have been hypothesized.

## Measures
### Valence and arousal

For each of the 4 pictures 1 item was used to assess valence (i.e., "Describe how unpleasant do you think this situation is") and 1 item to assess arousal (i.e., "Describe how physiologically arousing do you think this situation is"). Participants responded on 1 (not at all) to 9 (extremely) scales for both measures, and the scales were labeled only on point 1 and 9. In between no labels were given as this has been shown to be sufficient and an understandable scale labelling for different variables in previous studies (e.g., *Elliot et al., 2010*).

### Trait anxiety

Based on *Spielberger et al.*'s *(1983)* Trait Anxiety Inventory, the German 20-item measure by *Laux et al. (1981)* was used to assess trait anxiety (e.g., "I worry too much over something that really doesn't matter"; $\alpha = .87$). Participants responded on a 1 (almost never) to 4 (almost always) scale.

### Test anxiety

*Pekrun et al.*'s *(2004)* 12-item Test-Emotions Questionnaire (TEQ) was used to assess test anxiety before exam (e.g., "I worry whether I learned enough"; $\alpha = .75$) and during exam (e.g., "I feel panicky when writing an exam"; $\alpha = .77$). Participants responded on 1 (not at all true) to 5 (absolutely true) scales.

### Ethics statement

The research reported herein was conducted at the LMU Munich and was approved by the ethics committee of the Department of Psychology, LMU Munich, in accordance with the ethical standards of the American Psychological Association (APA). All participants gave written informed consent and were thoroughly debriefed. The individuals' consent was obtained after reading the instruction to the experiments. The experimenter asked for the participant's consent and emphasized that they will receive their credit also if they decided not to participate in this study. Participants were also told that they could stop and leave the experiment at any point of time. Data were stored and analyzed anonymously. This consent procedure has been approved by the ethics committee.

## RESULTS

### Arousal

A between-subjects analysis of variance (ANOVA) with color and context as between-subjects factors was conducted on arousal. Results revealed no significant two-way interaction on arousal, $F = 1.10$, $p > .250$. However, there was a significant main effect for color ($F(1,301) = 4.36$, $p = .038$, $\eta_p^2 = .01$, $d = .20$) and a significant main effect for context ($F(1,301) = 115.64$, $p < .001$, $\eta_p^2 = .28$, $d = 1.23$). Participants rated the red pictures ($M = 3.83$, $SD = 1.16$) more arousing than the blue ones ($M = 3.65$, $SD = 1.06$, see Fig. 1) and the test pictures ($M = 4.32$, $SD = 1.05$) more arousing than the neutral ones ($M = 3.17$, $SD = .84$).

### Valence

A further ANOVA with color and context as between-subjects factors was conducted on valence. There was a significant main effect of context ($F(1,301) = 325.48$, $p < .001$, $\eta_p^2 = .52$, $d = 2.05$). Participants rated the test pictures ($M = 4.49$, $SD = 1.03$) more negative than the neutral ones ($M = 2.47$, $SD = .95$). In addition, there was a significant main effect for color ($F(1,301) = 5.40$, $p = .021$, $\eta_p^2 = .02$). Participants rated the red pictures ($M = 3.57$, $SD = 1.42$) more negative than the blue ones ($M = 3.39$, $SD = 1.41$) and this main effect was moderated by a significant two-way interaction, $F(1,301) = 4.12$, $p = .043$, $\eta_p^2 = .01$, $d = .16$.

To determine the exact nature of this moderation we additionally ran Scheffe post-hoc tests within each context condition. In the test condition, participants rated the pictures more negatively in the red ($M = 4.75$, $SD = .86$) than in the blue ($M = 4.26$, $SD = 1.12$) condition, $p = .026$. In the neutral condition, no difference emerged between red and blue pictures, $p > .25$. Additional post-hoc tests within the color conditions revealed that in the red condition, participants rated the test pictures ($M = 4.75$, $SD = .86$) more negative
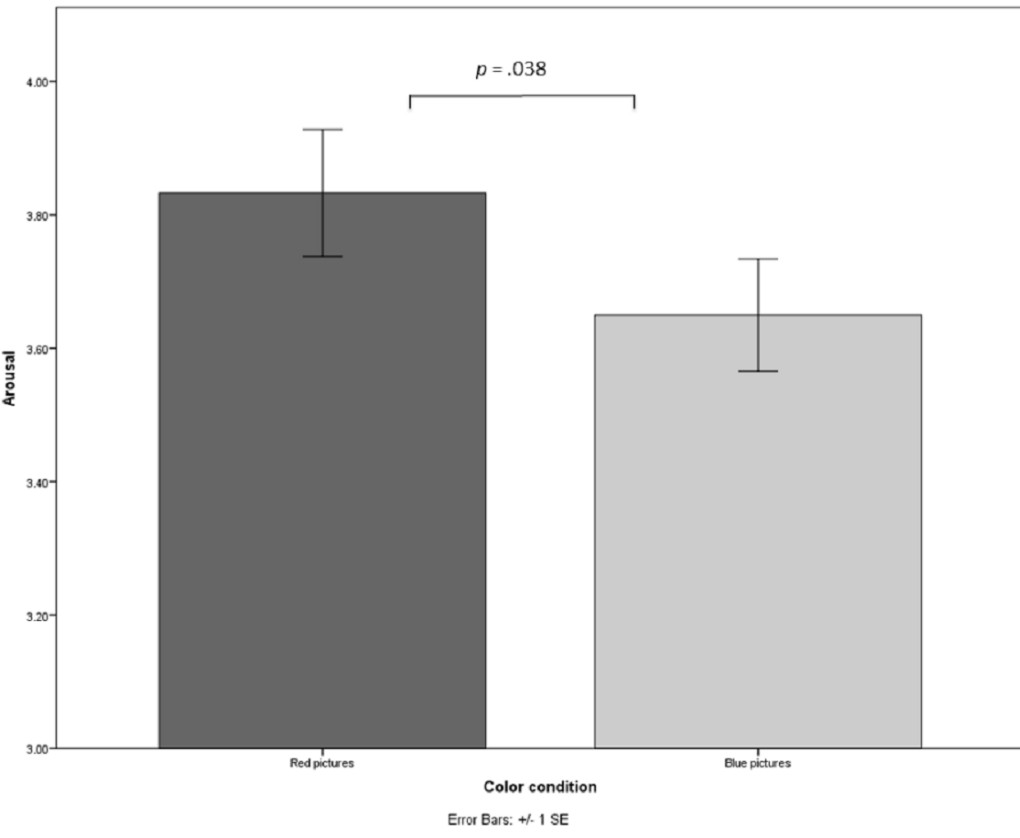

**Figure 1** **Arousal perceptions of red and blue pictures.**

[3] In the demographic section at the end of the experiment, we asked participants to rate how much their perception was affected by the colored frame on a 1 (not at all) to 9 (completely) scale. Subjective ratings showed that most of participants indicated that their judgement was not affected by color (MDN = 2.00, M = 2.33, SD = 1.72). This indicates that the majority of participants in our sample was convinced that color did not affect their behavior in the various tasks they performed. Hence, attributing the observed color effects to an explicit level should be treated with caution. In addition, we also asked our participants to recall the color of the frame. Only 53.77% could correctly remember the color. We speculate that this low recognition rate is another indicator of an incidental, implicit effect of color.

than the neutral ones ($M = 2.49$, $SD = .86$), $p < .001$. In the blue condition, participants rated the test pictures ($M = 4.26$, $SD = 1.12$) also more negative than the neutral ones ($M = 2.45$, $SD = 1.05$), $p < .001$, however to a lesser degree (see Fig. 2). [3]

Supplementary analyses controlling for both covariates, trait and test anxiety, showed that these measures of individual differences did not substantially moderate the interaction effect on valence reported above.

## DISCUSSION

The present study examined the influence of red on both arousal and valence perceptions of test-relevant and neutral stimuli. As mentioned before, arousal theorists, like *Wexner (1954)* and *Goldstein (1942)*, state that the color red per se is arousing. More precisely, they claim that the perception of red is associated with terms such as 'stimulating' as well as produces physiological reactions such as faster speed of movement, increased blood pressure, and frequency of eye blink. Research on the color-in-context theory in contrast, could show that context in which the color red is perceived influences its valence-related meaning and behavioral responses to it (e.g., *Elliot & Maier, 2012*; *Maier et al., 2009*).

With regard to the arousal theory, red was expected to raise the present study's arousal perceptions for both picture types (i.e., test-relevant and neutral ones), as red is supposed

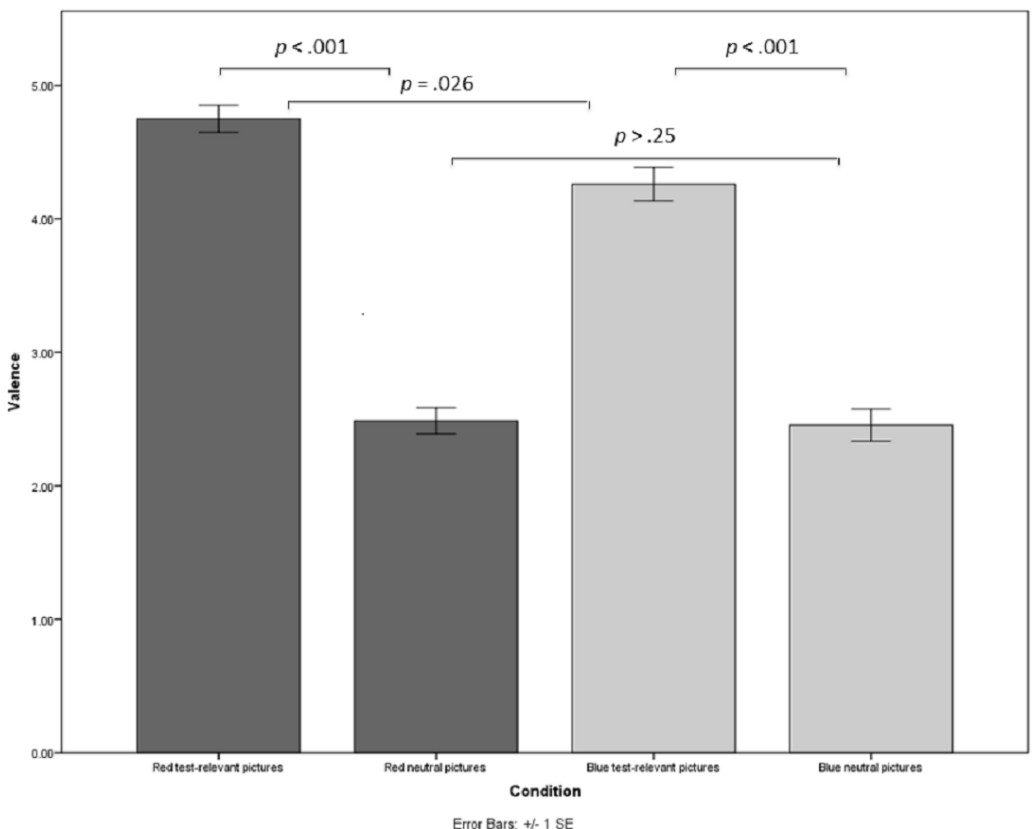

**Figure 2** Valence perceptions of red test-relevant, red neutral, blue test-relevant, and blue neutral pictures.

to be arousing independently of a given context (see for instance *Crowley, 1993*; *Gerard, 1958*; *Nakshian, 1964*; *Wilson, 1966*). However, with regard to the color-in-context theory red was expected to raise valence perceptions only for test-relevant pictures, as context is said to be the determining factor. Context dependent moderations of red have consistently been shown for valence related variations. As such, *Buechner et al. (2015)* could show that red increases the perceivers' valence dependent response tendencies, that is increased approach in case of a positive-appetitive stimulus configuration and increased avoidance in case of a negative-aversive one. Further, *Buechner et al. (2014)* could show that such context dependent red effects could be explained by attentional bias towards a specific stimulus. More precisely, red has been shown to accentuate the relevance of a goal-related stimulus and correspondingly intensify the perceivers' attentional reaction to it.

In line with these predictions the perception of red appears to be arousing, no matter if the stimuli is neutral or negative (i.e., test-relevant), but with regard to valence the perception of red intensifies the negativity only of aversive stimuli. Controlling for the covariates, trait and test anxiety, did not substantially moderate the interaction effect on valence. As such, independently of the trait and test anxiety of subjects, reddened test-relevant pictures were perceived as more negative than blue test-relevant ones as well as red neutral ones.

Previous research has shown that the recognition of facial and bodily expressions is influenced by the context (e.g., *De Gelder et al., 2006*; *Kret & De Gelder, 2010*; *Righart & de Gelder, 2006*). That is, the affective information provided by the surrounding scene in which a face/body appears may influence how it is encoded (*Righart & De Gelder, 2006*). Congruence between the valence of a facial or bodily expression with the valence of a context enhance the emotion perception. In the study presented herein, we used pictures that present congruent target-context information to make sure that the valence is best perceived.

Our results were in line with previous research on both the arousal theory of color (from a cognitive perspective) and the color-in-context theory, and contribute to the psychological literature by facing both theories within one study. However, the present study is limited in some senses that need to be taken into account for further research. As this study focuses on undergraduates only, further research is needed that concentrates on students or adults. Also, the color manipulation done by a frame raises questions regarding the degree to which the findings would generalize to other types of color presentation. A valuable follow up to this study would be a replication that focuses on another age sample and varies the color manipulation.

As mentioned above, an a priori power analysis was performed based on a small effect size ($d = .16$). We prefer to refrain from post hoc power analyses, as this procedure is known to be fundamentally flawed (*Hoenig & Heisey, 2001*). Nevertheless the central effects reported in our paper are indeed very small. Therefore, another preregistered and confirmatory study could help to clarify the robustness of the color effects. We would like to encourage other independent labs to replicate our findings. However, we still think that our results are worth to be reported since many psychological effects have been shown to be very small (30.44% yielding an $r$ of .10 or less; *Richard, Bond & Stokes-Zoota, 2003*). In addition, the predefined sample size based on high a priori power allowed us to interpret the results on a statistical level. The question about the practical impact is legitimate to ask but was not the primary goal of this study. Rather, we tried to test different theories of effects of the color red on arousal and valence perceptions.

The present research was designed to differentiate two color theories within one study. In our view we successfully managed this research goal with a highly powerful sample and carefully designed control conditions. We hope that the present results are useful for further research to extend this type of work.

### Funding

This research was supported by the German Research Foundation (Deutsche Forschungsgemeinschaft (http://www.dfg.de)) Grant # MA 2447/4-1. The funders had no role in study design, data collection and analysis, decision to publish, or preparation of the manuscript.

## Grant Disclosures

The following grant information was disclosed by the authors:
German Research Foundation: MA 2447/4-1.

## Competing Interests

The authors declare there are no competing interests.

## Author Contributions

- Vanessa L. Buechner conceived and designed the experiments, performed the experiments, analyzed the data, contributed reagents/materials/analysis tools, wrote the paper, prepared figures and/or tables, reviewed drafts of the paper.
- Markus A. Maier conceived and designed the experiments, analyzed the data, wrote the paper, reviewed drafts of the paper.

## Human Ethics

The following information was supplied relating to ethical approvals (i.e., approving body and any reference numbers):

Ethics committee of the Department of Psychology, LMU Munich.

The research reported herein was conducted at the LMU Munich and was approved by the ethics committee of the Department of Psychology, LMU Munich, in accordance with the ethical standards of the American Psychological Association (APA). All participants gave written informed consent and were thoroughly debriefed. The individuals' consent was obtained after reading the instruction to the experiments. The experimenter asked for the participant's consent and emphasized that they will receive their credit also if they decided not to participate in this study. Participants were also told that they could stop and leave the experiment at any point of time. Data were stored and analyzed anonymously. This consent procedure has been approved by the ethics committee.

## Data Availability

Buechner V. 2016. Data_Not always a matter of context: Direct effects of red on arousal but context-dependent moderations on valence. Open Science Framework. September 2. Available at osf.io/vzw5r.

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
