# Peer review of "Not always a matter of context: direct effects of red on arousal but context-dependent moderations on valence"

_PeerJ, doi:10.7717/peerj.2515_

## Round 0.1 · original submission · Major Revisions

Both reviewers are asking for additional information about the statistical approach to the data. Myself, as Academic Editor, fully agree on their concerns.

Reviewer 1 ·

Basic reporting

Introduction and discussion could profit from illustrating specific aspects of their study and the field in more detail:
First, the authors focus on specific conditions within the color-in-context field: The critical information in the study has already an affective value; in many color-in-context studies information gets its affective value via the color (e.g., less food is consumed in the context of red; food can, however, be generally seen as positive or palatable). Thus, the impact color can exert in their study is different from other studies: It slightly intensifies the meaning of the negative test items, but does not change it.
Second, it seems that the authors want to extend their previous research onto a different field, that is, from a romantic to an achievement context (or from a positive to a negative context). This could be made clear as well. Third, many studies in the color-in-context field highlight the automaticity of meaning activation. The authors used relatively explicit conditions in their study. Do they think explicit processing is needed for the observed results? Perhaps, explicit processing is only needed to observe the impact of arousal, but not valence? Fourth, the arousal theory and the color-in-context theory are based on very different levels of processing. Specifically, the arousal theory posits that red deserves its meaning from its wavelength characteristics (see l.118ff), while the color-in-context theory focuses on the affective meaning of red on a more cognitive level. From this perspective, it seems logical that the associated arousal does not change with context. The point of the two different levels could be clarified a bit more.

Experimental design

No comments

Validity of the findings

I am a bit concerned about the very small effect sizes associated with some relevant effects. For example, the main effect of color on arousal is only associated with η²p=.01. This is a very small effect (replication of this effect size would need approx. 570 participants, calculated with GPower). Thus, I wonder about the practical impact of this effect? The same holds for the two-way interaction of context x color, η²p=.01. Thus, it seems that context was the defining factor in their study, and color almost negligible. This should be made clear. The authors should be more cautious in their conclusions.

Additional comments

Minor points:

l.127 „these set of circumstance“ should be „these circumstances“ or „this set of“ or „these sets“ ?

l. 159 sentence could profit from commata, to enhance readibility

l. 167: Sentence seems incomplete: „The use of picture stimuli was limited to color research in romantic settings, but never in achievement contexts“. But has so far not been employed in…?

l.188: I don’t understand the comment in parentheses („reason for its inclusion“). What was included? What do the authors want to say?
l.260 „fully moderated“ , „fully“ should be deleted, what would partial moderation be?

·

Basic reporting

Some minor comments / suggestions:
• line 118: “explicit and implicit measurement tools” could be misleading to some readers. Please refer to examples from the above section which is an explicit or implicit measurement
• Section Materials & Methods (PeerJ template) is divided in Method (line 174) and Measures (215) in this paper. I recommend restructuring this section to fit the PeerJ structure.

Experimental design

Some minor comments / suggestions:
• lines 191-207: Please specify more detailed the material. For example, it could be interesting for replication attempts to know the sizes of the sheets on which the pictures were printed, the size of the pictures itself, and the way the pictures were framed with red or blue (for example, were these pictures printed on a red / blue sheet, or only surrounded by a rather small frame?)
• lines 223-230: Why did the authors include trait and test anxiety? As these variables were also included as covariates in the results section, it could be interesting to know the rationale for measuring these variables.

Validity of the findings

Some minor comments / suggestions:
• Results section: the authors report effect size estimates to determine the sample size (d = .16; footnote 1). In the reports section, they include partial Eta2 as effect size measures. I recommend including also Cohen’s d in the results section. Thus, readers can see on first glance the effect size and compare it with the a priori estimated effect size. Including Cohen’s d (for example) would also enable the reader to follow the authors reasoning on lines 267-270 and Figure 2.
• Results section: I wonder why the authors included trait and test anxiety only on the valence, but not on the arousal ratings (lines 272-274).

Additional comments

I really enjoyed this very clear and precise written paper. All minor comments / suggestions could be very easy implemented by the authors. Thus, I recommend a minor revision.

---

## Round 0.2 · accepted · Accept

Reviewer 1 suggested a minor change in the text which you can fix in production.

Reviewer 1 ·

Basic reporting

No comments

Experimental design

No comments

Validity of the findings

No comments

Additional comments

The authors satisfactorily responded to all my questions and comments. I recommend a publication of the article.

MInor comments/suggestions:
In re-reading the article, I really had the impression that the process(es) underlying the effect of red on arousal are not entirely clear. It seems worthwile to investigate this topic in more detail. Specifically, it seems that the arousal-related processes could run on several levels: a physiological level triggered by wavelength, an attention-related level and on a more semantic, association-related level. Thus, further dependent variables would be interesting. As mentioned by Reviewer 3, a physiological measures (i.e., skin conductance) would be interesting. To study the effects of attention, eye tracking might be useful. Fixation durations might be longer for red-framed compared to other objects. Finally, it could also be that the effect is related to the subjective impression, regardless of attention and physiology. The level on which the effect is based might also explain why no moderation with anxiety was observed. Perhaps, arousal resulting from general anxiety does not influence the effect, because the effect is based on different processes. Perhaps, the effect of red resulting from a semantic level could be context-sensitive for arousal as well. Testing this process question might be an interesting avenue for future research.

If the reference in l.147 is deleted. Then, the publication year should be mentioned in the following line.

·

Basic reporting

No comments

Experimental design

No comments

Validity of the findings

No comments

Additional comments

After carefully reading all reviews of the first submission, the rebuttal letter, and the new submission, I have no further comments and recommend accepting the article in its current form.